# Robotics in Dentistry: A Narrative Review

**DOI:** 10.3390/dj11030062

**Published:** 2023-02-24

**Authors:** Lipei Liu, Megumi Watanabe, Tetsuo Ichikawa

**Affiliations:** Department of Prosthodontics and Oral Rehabilitation, Graduate School of Biomedical Sciences, Tokushima University, 3-18-15 Kuramoto, Tokushima 770-8504, Japan

**Keywords:** robotics, dentistry, prosthodontics, oral implantology, oral surgery, orthodontics

## Abstract

Background: Robotics is progressing rapidly. The aim of this study was to provide a comprehensive overview of the basic and applied research status of robotics in dentistry and discusses its development and application prospects in several major professional fields of dentistry. Methods: A literature search was conducted on databases: MEDLINE, IEEE and Cochrane Library, using MeSH terms: [“robotics” and “dentistry”]. Result: Forty-nine articles were eventually selected according to certain inclusion criteria. There were 12 studies on prosthodontics, reaching 24%; 11 studies were on dental implantology, accounting for 23%. Scholars from China published the most articles, followed by Japan and the United States. The number of articles published between 2011 and 2015 was the largest. Conclusions: With the advancement of science and technology, the applications of robots in dental medicine has promoted the development of intelligent, precise, and minimally invasive dental treatments. Currently, robots are used in basic and applied research in various specialized fields of dentistry. Automatic tooth-crown-preparation robots, tooth-arrangement robots, drilling robots, and orthodontic archwire-bending robots that meet clinical requirements have been developed. We believe that in the near future, robots will change the existing dental treatment model and guide new directions for further development.

## 1. Introduction

“Robot” is a new term that emerged in the 20th century. In 1920, the Czech writer Karel Čapek published the science fiction script *Rossum’s Omnipotent Robots*, in which the word “robot” was first coined from the Czech word “Robota,” with a meaning similar to “labor” or “drudgery” [1]. At the first robotics conference held in Japan in 1967, Masahiro Mori put forward a representative definition of robots: “A robot is a flexible machine with seven characteristics: mobility, individuality, intelligence, versatility, semi-mechanical, semi-human, automatic, and a slave.” The American National Standards Institute defines a “robot” as a mechanical device that can be programmed and can perform certain operations and mobile tasks under automatic control.

With the rapid development of modern science and technology worldwide, robotics has become a popular research area and social concern. Robotics has been applied in many fields such as machinery, electronics, aerospace, and medicine. Among these, the application of robotics in medicine has attracted increasing attention. Medical robots are divided into macro-, micro-medical, and bio-robots. Macrorobots mainly include rehabilitation robots (such as home daily-care robots, and smart wheelchairs) and surgical robots (brain-surgery and eye-surgery robots), which also include minimally invasive surgical robots and medical endoscopic devices. Bio-robots are medical robots that can perceive, think, and judge as humans do [2].

The successful application of medical robots has also garnered enthusiasm for research on robotics in dentistry, which breaks through the previous oral diagnosis and treatment models and promotes a new avenue of technological innovation. This article provides a comprehensive overview of the basic and applied research status of robotics in dentistry and discusses its development and application prospects in several major professional fields of dentistry: prosthodontics, oral implantology, oral surgery, orthodontics, and endodontics. This article describes fully automatic robotic systems and semi-automatic human–robot collaborative robotic systems, but does not include robot-assisted systems that are entirely operated by a surgeon, such as the da Vinci Surgical System. We believe that, with the deepening of the application and research of robots in dental treatment, research, and education, their development can guide new directions for the future of dentistry.

## 2. Materials and Methods

The literature search was based on the MEDLINE, IEEE, and Cochrane Library databases. The last search was conducted on 28 December 2022. The authors used the following combination of MeSH terms: [“robotics” and “dentistry”]. The authors also manually searched the reference lists to identify relevant papers that were not captured through the search strategy. Articles were selected according to the following criteria (Figure 1).

The inclusion criteria were full-text articles exclusively published in English language, and the use of robotics to assist in treatments within the fields of dentistry, including but not limited to prosthodontics, orthodontics, implants, endodontics, oral surgery, and others. The exclusion criteria were investigations regarding the use of robots as simulated patients or in conversational applications, designs of robots incapable of performing actions autonomously, publications including essays, conference abstracts, letters, and commentaries, and works published before the year 2000.

In the instance where multiple publications from a single study population were uncovered, the authors included only the one that provided the most comprehensive information regarding the results. Eligible studies were identified through a consensus process, and Table 1 highlights representative articles that demonstrate recent and pioneering advancements in robotics across various countries and diverse domains of dentistry.

## 3. Results

277 articles were generated in the first search, after screening at the abstract and full-text level; finally, 49 articles were selected. After a rough classification, we found that among all articles, there were 12 studies on prosthodontics, which is the largest proportion of the dentistry field, reaching 24%. The second most common was dental implantology, with 11 studies accounting for 23% (Figure 2a). Regarding country distribution, scholars from China published the most articles, followed by Japan and the United States (Figure 2b). From the perspective of publication time, after 2000, the overall trend in the number of articles on dentistry robots has been on the rise, with the largest number of articles having been published between 2011 and 2015 (Figure 2c).

### 3.1. Prosthodontics

The applications of robots in prosthodontics can be divided into the following three parts: Recent research has mainly focused on the development of tooth-preparation and tooth-arrangement robots, and their high standards, high intelligence and precision, and other advantages have become the development direction of intelligent prosthodontic treatment.

#### 3.1.1. Tooth Preparation

Tooth preparation is the process of quantitative preparation and formation of hard tissue in diseased teeth. Currently, tooth preparation is performed by using high-speed dental handpieces. However, many associated risks exist. Dentists need to avoid damaging the soft tissues in the narrow space of the oral cavity and attempt to achieve accurate tooth preparation standards simultaneously. Excessive or insufficient preparation may cause problems such as thermal hypersensitivity and pulp inflammation, which may cause soft tissue damage, thereby affecting the quality of restoration.

Compared with traditional turbine-driven drills, laser ablation of hard tissues for tooth crown preparation is considered safer and more comfortable because it generates less noise and vibrations. In 2013, a microrobot device called LaserBot was launched. This robotic device achieved precise three-dimensional (3D) motion control of a femtosecond laser beam to realize clinical tooth-crown preparation. Moreover, the device could be mounted on any tooth owing to its small size [9]. A conceptual illustration of the automatic crown preparation system is presented in Figure 3. Some researchers have conducted experiments on wax, resin, and teeth using a system that combines robotic and laser technology to achieve automatic 3D tooth ablation. The results proved that this robotic system could meet the requirements for dental crown preparation in typical dental operations. However, the ablation time when using this robotic system was much longer than expected [18]. The same team developed a precisely controlled ultra-short pulse-laser automatic tooth-preparation robot device using 3D motion planning software. In addition to ensuring the accuracy and feasibility of tooth preparation, the average preparation time of newly extracted human intact first molars has been shortened to 17 min [14].

Otani et al. conducted experiments to compare the accuracy and precision of an automatic robotic tooth-preparation system for porcelain-laminated veneers and conventional freehand tooth preparation. Comprehensive data showed that the robotic tooth-preparation system had similar accuracy to the conventional freehand tooth-preparation system, but the precision was lower [19].

#### 3.1.2. Tooth Arrangement

Traditional complete dentures are fabricated manually, and the key step is to arrange the artificial teeth into a dental pad in the appropriate position and direction. Senior dentists and technicians generally require high skills to complete this process. With the development of computer-aided design/computer-aided manufacturing (CAD/CAM) technology, virtual artificial tooth-arrangement systems have gradually been developed. In recent years, it has become possible to use 3D digital technology to achieve more complex jaw position recording and balanced tooth arrangement. In the software, the entire dentition, local dentition, and single tooth can be moved freely in the horizontal, sagittal, and coronal planes, and the dentition radians can be changed locally to adapt to the shape of the dental arch.

The data stored in the CAD process is further processed by CAM into instructions for operating and controlling production machinery. At the laboratory design level, CAD/CAM programs eliminate differences in experience between dental technicians; however, at the manufacturing level, robotic systems with sophisticated operational capabilities are required to minimize human error, harmonize design standards, and improve production speed and efficiency.

Research on the operation of tooth arrangement using robots is currently in progress. Zhang et al. developed a robotic manufacturing system to arrange teeth for complete dentures and successfully obtained an ideal dental arch for the patient. First, the patient’s jaw arch parameters were obtained, and control data were created using tooth-arrangement software. The robot then grabbed and assembled the intermediate blocks, tooth-arrangement helper, and artificial teeth. Finally, wax was poured into the tooth-arrangement helper to obtain a fixed tooth arch, and the complete denture was transformed [20]. The same team achieved significant progress in the development of tooth-arrangement robots. In 2008, they proposed a tooth-arrangement robot composed of five-degree of freedom (DOF) mechanisms in series and parallel structures and established kinematic equations [21]. A method of tooth arrangement using a multimanipulator was proposed in 2009 [22]. The method of using high-resolution timing control pulses to achieve the high-precision coordinated motion control of a tooth-arrangement robot arch generator was studied in 2010 [23].

In the theoretical section, Zhang et al. made contributions. To solve the time-consuming and low-precision problems of traditional tooth-arrangement methods, they proposed the concept of a professional and miniature Cartesian coordinate-type tooth-arrangement robot [24]. Based on the analysis of the motion of the dental arch generator, they determined the objective function, multivariate design, and constraint function of the control point optimization and optimized its control points [25]. Based on a mathematical model, they investigated the automatic acquisition of dental arches and implemented motion planning and the synchronized control of the dental arch generator [7].

#### 3.1.3. Articulation

After the artificial teeth are arranged in the CAD software, the occlusal relationship can be adjusted by simulating jaw opening and closing movements and forward/backward and lateral movements using the virtual articulator. Robotic articulators are being studied in addition to virtual articulators.

A new type of robotic articulator uses a precise six-axis micropositioning stage to reproduce the patient’s functional mandibular movement with six DOF. Using this articulator system, a full veneer crown restoration is fabricated without the need for intraoral occlusal adjustments to the setup. Because the articulator can accurately reproduce dynamic jaw movements during functional jaw movements, the system has the potential to improve the accuracy of denture occlusion. However, since only one patient has been examined, further research is needed to evaluate this technique [3].

### 3.2. Oral Implantology

With the continuous development of implant technology, implants have gradually become the first choice for the restoration of missing teeth. However, the accurate insertion of the implant into the patient’s jaw, which is key to the success of dental implants, requires high anatomical accuracy. Improper drilling positioning can cause serious bleeding, nerve damage, and other accidents. Therefore, robots are expected to provide more reliable and successful dental implant results owing to their accuracy and stability. As early as the 1990s, many medical research teams used interactive computer applications, including hardware and software, as aids in implant planning. Around the 2000s, sensors were used in robotics to automatically monitor surgical procedures, expanding their applicability from presurgical plans to mainstream surgery.

Robotic applications in implantology can be broadly classified into robot-assisted and fully automated implantation robots. Yomi, developed by Neocis in the United States, is a commercial robot-assisted dental-surgery system. The operating arm of the Yomi system assists with automatic positioning by connecting the coordinate measuring machine arm to the patient’s teeth. It was approved by the U.S. Food and Drug Administration in 2017 and has shown impressive clinical results [26]. In the same year, Professor Zhao Yiming of the Air Force Medical University in China developed a fully automatic optical navigation system that could drill and install dental implants according to a preoperative plan. This system was used to complete the immediate implant placement of two anterior teeth in a female patient in 2017, and the results of the surgery were expected. In March 2022, the robot system took only 20 min to complete another accurate anterior tooth implant surgery in Beijing. The thickness of the patient’s alveolar bone was only 6 mm, and that of the implant was 4 mm. In the past, dental surgeons needed to cut the gingival mucosa before placing an implant. However, the robotic system can perform minimally invasive surgery by drilling directly into the gingival mucosa, significantly shortening the surgical and wound recovery time. During drilling, the robot can also follow the movement of the patient to calibrate the position, reducing human errors to a lower level, making it more accurate and safer.

In addition to the assisted or automatic system, a team has constructed a human–machine collaborative dental implant system to fully utilize the advantages of both the surgeon’s experience and robot’s stability [27]. The design and simulation of a human–robot cooperative manipulator for implant surgery have also been studied [28]. To overcome the technical difficulties associated with oral implant treatment, a computer-aided oral implant system was introduced, including framing, modeling, preoperative 3D planning, registration, optical tracking, and real-time navigation systems [29]. The construction of a computer-aided system combining preoperative planning and surgical navigation [4] and a homogeneous transformation-algorithm-based navigation system for implant surgery [30] has also been proposed. An image-guided robotic system for dental implants was proposed in 2011 [31]. Three years later, an improved version of this robotic system, which allowed for more precise drilling of complex types of implants, was introduced [10].

Onishi et al. proposed a telerobotic-assisted drilling system. A reaction force observer was used to measure the cutting force, which can be transmitted from the material to the surgeon [6]. Onishi et al. also applied a navigation system based on stereo vision to a three-DOF implant-surgery-assisting robot, allowing the surgeon to operate the manipulator freely away from the destination. Simultaneously, the manipulator adjusts the force when approaching the destination to correctly converge and avoid collisions [32]. A method for simulating the cutting force response and predicting the vibration during bone drilling was proposed by the same team. This method uses the relationship between the parameters and the cutting material density to simulate the equations of the cutting force and CT number. It allows dental students to learn and practice procedures, such as cutting jaws [33].

In conclusion, the role of intelligent robots in oral implantation mainly includes: (1) preoperative digital 3D scanning of the implant site and imaging data collection/diagnosis analysis; (2) digital implant surgery plan design; and (3) real-time navigation and automatic drilling during the operation to improve the accuracy of dental implant surgery, reduce surgical trauma, and shorten the operation time.

### 3.3. Oral Surgery

With the development of computer-assisted surgery, the preoperative design of maxillofacial surgery has continued to improve. However, solving the real-time accuracy and stability of drilling and cutting procedures remains a challenge.

Mandibular reconstruction is a complex and challenging process. Conventional reconstructive surgery generally requires two groups of surgeons working for a minimum of 7 h. Considering its limited manual accuracy and human resources, robot-assisted surgery has emerged as an alternative. Experiments on osteotomies performed by a robot (KUKA, Augsburg, Germany) in a pre-programmed manner for fibula free-flap mandible reconstruction have been reported, and this method showed a high degree of accuracy [15]. Another study compared the feasibility and accuracy of mandibular reconstruction using an automatic three-arm robot system and freehand technology under the guidance of a computer-aided navigation system. The robotic system functioned normally and stably and successfully performed mandibular reconstructions in both phantom studies and animal experiments [16]. Sun et al. conducted an experiment on fully automatic robot-assisted surgery for mandibular angle split osteotomy on five beagles. The results revealed an acceptable accuracy error and no postoperative complications occurred [34].

In orthognathic surgery, a robotic system has been developed to assist in bone segment repositioning. The system consists of an arm with six DOF, a robot motion controller, and a PC. The tracking tools at the end effector and patient splint helped to reposition the phantom maxillary complex around the tool center point of the maxillary incisor and mandible [35].

An autonomous oral and maxillofacial surgery system with the assistance and supervision of a surgeon was developed (Figure 4). The system was used to conduct drilling experiments on five 3D-printed mandibular models. The results showed that the system could successfully guide the robot to complete the operation, regardless of the mandibular posture, with acceptable accuracy [36].

The role of robots in oral and maxillofacial surgery mainly includes: (1) the acquisition and reconstruction of 3D image data of the oral and maxillofacial before the operation, analysis of the characteristics of the lesion, and design of a targeted operation plan; and (2) the accurate segmentation, reshaping, displacing, and fixing of the craniofacial bone according to the surgical plan. Robotics has been successfully used in oral and maxillofacial surgery, and robots for special operations in oral and maxillofacial surgery, such as velopharyngeal surgery, are also being developed.

### 3.4. Orthodontics

Currently, robots can be used in orthodontics for clinical diagnosis and the preparation of treatment plans; however, the most widely used aspect is archwire bending. Archwire bending is one of the most critical components of orthodontic treatment; however, owing to the high stiffness and superelasticity of orthodontic wires, this is not a simple task. The traditional manual method for obtaining the formed archwire curve randomly introduces errors caused by human factors.

Zhang et al. designed a novel robotic system for bending archwires. They studied the structure and components of the archwire bending robot system and established a coordinate system. A preliminary orthodontic-wire-bending experiment verified the feasibility of orthodontic wire bending using a robot system [37]. The same group also proposed a quantitative model of orthodontic wires with canine eminence. Based on the bending process analysis of orthodontic wires, they designed the overall structure, bend die, and archwire-supporting part of the archwire-bending robot [8]. Deng et al. designed an adaptive sampling-based bending planner with a collision checker in a time-varying environment and adopted a bending-control strategy to eliminate the spring-back effect of the archwire and bending-point deviation during the bending process. The system is illustrated in Figure 5 [38]. A year later, they added an ROS-integrated control system to an orthodontic archwire-bending robot [39].

A system called “LAMDA (Lingual Archwire Manufacturing and Design Aid)” has been introduced. This system contains a heater that can raise the temperature of a nickel–titanium archwire to 600 °F and bend it within 6 min. Blind evaluation of the archwires manually bent by 15 lingual orthodontic specialists and the archwires bent by the LAMDA system showed that those designed and manufactured using the LAMDA system had a higher score [40]. One study used the American Board of Orthodontics casting/radiology evaluation (CRE) to compare 63 patients receiving traditional manual wire-bending treatment with 69 receiving the SureSmile2 (SS) method performed by the same orthodontist. Compared with those receiving conventional treatment, the SS patients had lower CRE scores in first-order alignment and rotation and interproximal space closure; however, they had poorer second-order root alignment scores. Additionally, the treatment time for SS has been significantly shortened [41].

Orthodontic archwire-bending robots have the advantages of a simple structure, low cost, and can bend various types of archwires. This reduces the labor intensity of the doctor, prevents the fatigue fracture of the archwire caused by repeated bending, and improves treatment efficiency.

### 3.5. Endodontics

In root canal therapy, there are risks of a K-shaped file or rotary file breaking in the root canal, causing root perforation. A robot called the “Omni Phantom,” with a haptic virtual reality simulator has been developed to help users efficiently train in endodontic procedures. Using a simulated K-file, the user can experience the process of burring the enamel and dentin and cleaning the inner surface of the root canal [11]. Razavi et al. proposed a new approach to calculate the feedback force, and the speed of haptic loop execution increased approximately eight times in the process of a haptics-based tooth-drilling simulation [12].

Hong Seok at Columbia University proposed a project called “The Advanced Endodontic Development.” This project aimed to develop an intelligent microrobot that can perform endodontic treatment automatically. Dong et al. discussed the mechanical design and manufacturing of this robot and its innovations. According to the conceptual design, the robot will be mounted on several teeth in the patient’s mouth, and a 3D tooth model will be built using two-dimensional radiographic images. A prescription system will design the treatment procedures, and the microrobot will perform automated root canal drilling and filling [42].

In addition to the robotics used in endodontic treatment, the development of micro-endodontic robots can overcome the limitations of traditional treatment, such as the insufficient mouth opening, and can provide patients with more safe, accurate, and reliable root canal treatment. However, further research is still required for the design and manufacture of microsensors and actuators.

### 3.6. Others

#### 3.6.1. Mastication

Masticatory robots are robots, devices, or simulators that can simulate the human chewing motion. It can be used in dentistry, food science (evaluation of food texture characteristics and food-chewing dynamics), and biomechanics (analysis of mandibular joint force and stress distribution). It can provide jaw movement recordings for patients requiring prosthetic rehabilitation. It can also be used in the research and diagnosis of temporomandibular joint (TMJ) diseases or the study of mandibular kinematics during speech.

For the theoretical studies, in 2003, Usui et al. analyzed the mechanical strain distribution on the mandibular bone surface with a computational controlled mastication robot system [43]. In 2005, Wang et al. described the design of a novel biomimetic chewing robot, including its motion, force, control, and mechanical designs, and conducted some initial experiments on motion tracking [13]. To better determine the interactive loading between a dental prosthesis and the host mandible, Tahir et al. developed a robotic mastication simulator that incorporated a Stewart parallel kinematic mechanism and successfully replicated the human mastication force cycle [44].

In studies in clinical settings, Conserva et al. verified that a mechanical chewing simulator was able to reproduce mandibular movements and recognize the stresses delivered by three different restorative materials (acrylic resin, composite resin, and glass-ceramic) to the simulated bone-implant interface [45]. Conserva et al. also analyzed the force transmission of four different occlusal materials to a simulated peri-implant bone using a masticatory robot [46]. Raabe et al. developed a new in vitro wear simulator to replicate chewing movements and tested the wear resistance of new dental restorative materials [47,48]. Wen et al. designed a redundantly actuated humanoid chewing robot that was the first of this kind with two higher kinematic pairs to mimic the human TMJ. The experiments verified that the robot’s performance could meet the workspace requirements and accomplish functional movements similar to those of human beings [49]. Massimo et al. proposed a system consisting of a jaw movement analyzer and a robotic articulator called bionic jaw motion (Bionic Technology, Vercelli, Italy), which could quickly record individual mandibular motions, and analyze the recorded data on the robot. The upper arm reproduces the lower jaw movement, which overcomes many limitations of the traditional pantograph–individual articulator system [50].

#### 3.6.2. Tooth Cleaning

A six-axis robot that was programed with individual clinical toothbrushing programs performed toothbrushing on artificial teeth, which were covered with a plaque-simulating substrate. The total plaque removal efficacy of robot toothbrushing was significantly higher than that of clinical toothbrushing, and this new robot can be used for the laboratory testing of tooth cleaning (Figure 6) [51].

Sakaeda et al. developed an automatic tooth-cleaning mouthpiece robot to help elderly or handicapped individuals who cannot brush their teeth by themselves. The robot system consists of a tooth-cleaning module, flexible arm, water supply system, and safety system. With this safety system, the brushing action automatically stops if the user lifts his/her jaw from the jaw supporter, thereby avoiding the risk of lung inflammation caused by aspiration [50].

#### 3.6.3. Rehabilitation

An oral rehabilitation robot called Waseda–Asahi Oral Rehabilitation Robot No. 1 has been developed to conduct masseter and temporal muscle massage to ameliorate myofascial pain caused by oral and maxillofacial problems such as temporomandibular disorders (TMD) and dysphagia. A clinical trial of this robot in patients with TMD associated with myofascial pain has confirmed its safety and effectiveness and determined suitable conditions under which it could be used to administer massages [5].

## 4. Discussion

Although there have been successful cases of robot application in dental treatment, such as oral implantology and oral surgery, the operation process requires dentists to set the program and input the corresponding data on the basis of early judgment before the robot can complete the steps. During the operation, dentists are also required to monitor the process in real time. In case of data error, circuit interruption and other unexpected conditions, the consequences would be unimaginable without the help of dentists. On the other hand, for some complex problems, robots are still unable to solve them. For example, some implant patients need bone grafts, and these operations are difficult to be completed by robots.

At present, robots can participate in the oral treatment process moderately, reducing the working time of dentists, which also improves efficiency and reduces labor costs to a certain extent. However, compared with dental robots, excellent dentists are still indispensable. The new robot technology requires dentists to have not only traditional treatment technologies, but also digital technology, mechanical engineering technology and other composite skills. Only by increasing the training of dentists and improving the technical level of dentists can we fundamentally solve the problem.

The intervention of robotic technology in dentistry has the potential to provide improved and precise treatment in a shorter time and has a good quality of work. Beneficial explorations have been performed on the application of robotics in oral sciences, automatic tooth-crown-preparation robots, tooth-arrangement robots, drilling robots, and orthodontic archwire-bending robots that meet clinical requirements have been developed. However, these robots are still in the stage of theoretical research and preliminary experiments and require professionally trained and experienced operators; thus, they have not been widely used in clinical practice. Additionally, some highly repetitive and time-consuming operations in the oral cavity lack the application of robotic technology, such as basic periodontal treatment, the orthodontic bonding of brackets, archwire augmentation and restoration, and the removal of broken files in the teeth. Robotics may have broad prospects in these areas. Dental robotics is currently in the transition phase from computer-assisted operations to fully autonomous technology. Consequently, relevant research publications and case reports are limited. If we expect further standardization, industrialization, intelligentization, and wide-application in the daily teaching and clinical practice of dental robotics technology, the development of new structures, sensors, control theories and other related technologies and theories is essential. At the same time, multidisciplinary joint research such as stomatology, artificial intelligence, and mechanics should be promoted.

Currently, there is a burgeoning development of artificial intelligence (AI) technology for voice commands, exemplified by the advanced system, DEXvoice (DEXIS LLC, Alpharetta, GA, USA), which enables the effortless retrieval of X-rays, patient records, and charts by dentists, without requiring them to remove their gloves or engage with the computer manually. In addition, the AI company ParallelDots has introduced a cloud-based app that can identify dental cavities on dental X-rays. The integration of these cutting-edge technologies into dental robots may greatly enhance their capacity as effective assistants.

Despite the undeniable advantages of dental robots, which can operate with high efficiency and precision, several critical issues concerning security, privacy, and ethics must be taken into consideration. The first and foremost priority must always be the protection of patients’ safety, which forms the primary principle of diagnosis and treatment. In situations where the data is insufficient or inaccurate, there is a risk of errors in the judgment and execution results of the robots, thus potentially putting the patients in harm’s way. Secondly, the issue of privacy looms large. The application of AI technologies such as deep learning demands large amounts of medical data and information, and the inadvertent leakage of private information can have far-reaching negative consequences for patients, dentists, and institutions.

Furthermore, the ethical dilemmas posed by the use of robots to replace humans in diagnosis and treatment, the criteria for determining when robots should be employed in dental diagnosis and treatment, and the determination of liability in the event of medical accidents are all significant challenges that must be confronted. Consequently, a more comprehensive and standardized legal system is imperative for the effective application of robotics in the field of dentistry.

## 5. Conclusions

As an extension of the human hand and eye functions, the use of robotics in dentistry serves as a supplement to the limitations and inadequacies of manual operations, providing refined and precise movements beyond the capabilities of the human hand. As the integration of robotic technology into dentistry progresses, it will bring about a transformation in the conventional modus operandi and ambiance within the field, although it will not result in a widespread displacement of dental practitioners. In the future, the incorporation of advanced medical technology, including intelligent medical and service robots, will promote the provision of more accurate, efficient, and accessible dental services, as well as a more hospitable clinical environment for both dentists and patients alike.

## Figures and Tables

**Figure 1 dentistry-11-00062-f001:**
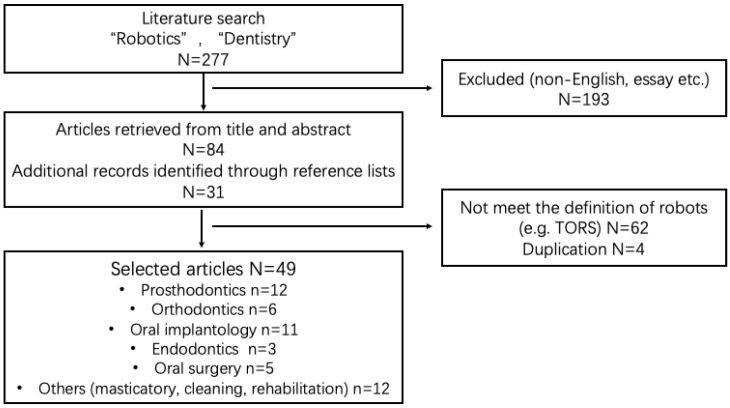
Flow chat for screening and selection of articles.

**Figure 2 dentistry-11-00062-f002:**
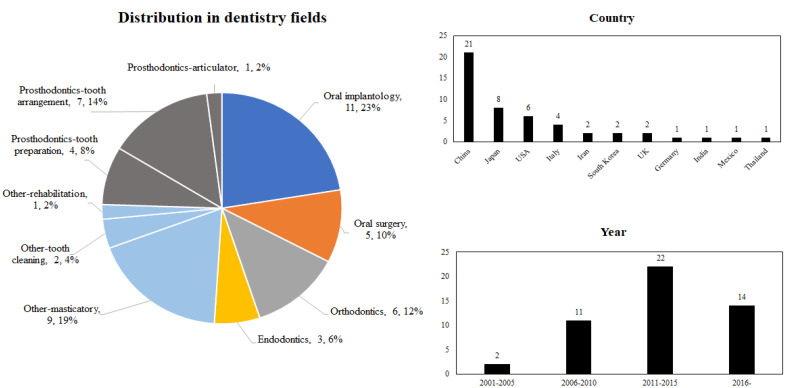
(**a**) Dentistry field distribution of selected articles; (**b**) Country distribution of selected articles; (**c**) Publication time distribution of selected articles.

**Figure 3 dentistry-11-00062-f003:**
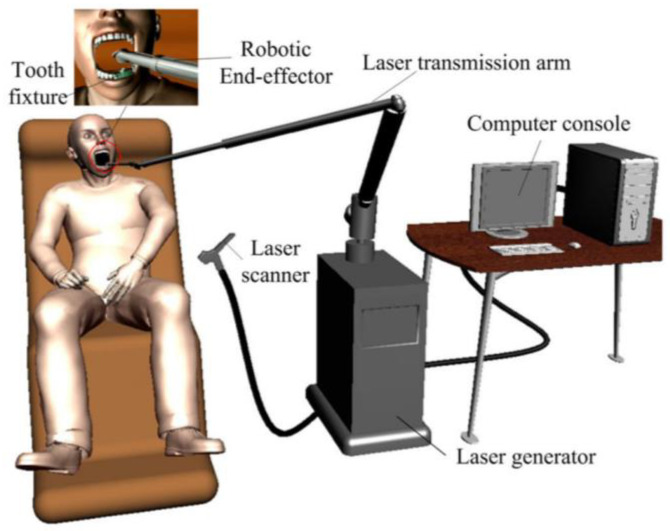
Conceptual illustration of the automatic crown preparation system [9].

**Figure 4 dentistry-11-00062-f004:**
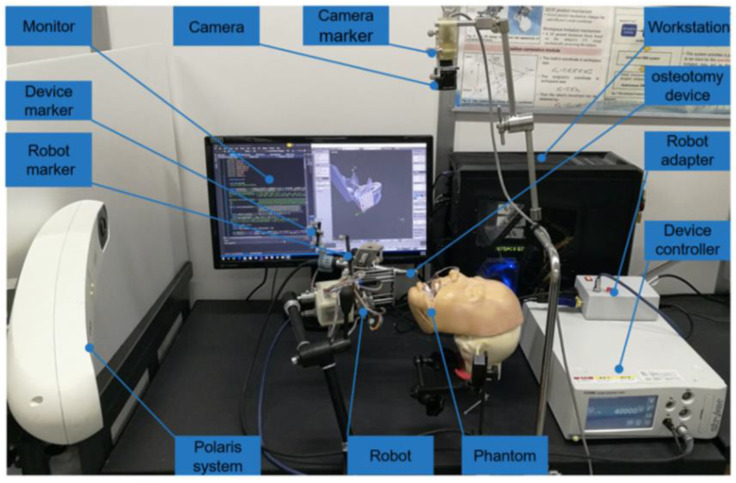
Overview of the autonomous surgical system for oral and maxillofacial surgery [36].

**Figure 5 dentistry-11-00062-f005:**
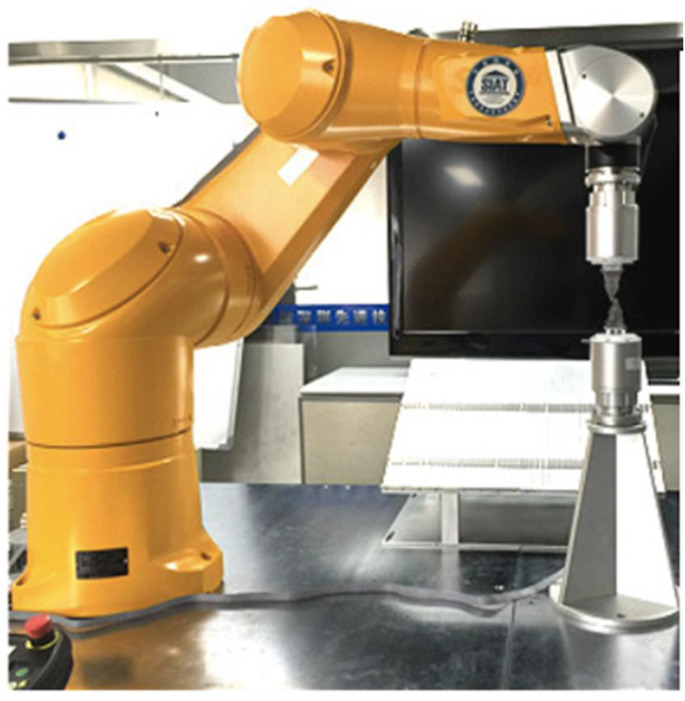
Novel robot system for orthodontic archwire bending [38].

**Figure 6 dentistry-11-00062-f006:**
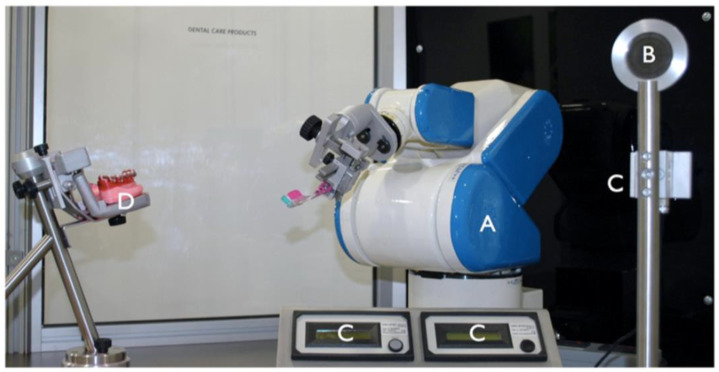
Overview of the toothbrushing simulation unit. A: six-axis robot. B: Calibrating graticule. C: Two shields for calibrating the brushing force. D: Mounting plate for mandibular typodont dentition 38–50 [51].

**Table 1 dentistry-11-00062-t001:** The characteristics of the representative articles in different dentistry fields.

Authors	Year	Country	Robot System	Dentistry Field	Outcome	Adventages	Reference
Nishigawa K, et al.	2007	Japan	A novel robotic articulator that reproduced a six-degree-of-freedom jaw movement	Prosthodontics-articulator	This articulator could perform a precise reproduction of the dynamic jaw motion during the functional jaw movement.	This system has potential to improve accuracy of the prosthetic teeth occlusion.	[3]
Zheng G, et al.	2007	China	A computer assisted system in dental implantology	Oral implantology	The preliminary experiment shows that the system is accepted in terms of both efficiency and the accuracy.	It is going to be test in clinical studies to improve accuracy and incorporatemore features that the dentists demandand.	[4]
Ariji Y, et al.	2009	Japan	A specially fabricated robot for masseter and temporal muscle massage	Other-Rehabilitation	The massage treatment was very effective for most patients.	The robot may constitute a useful tool for treating TMJ dysfunction associated with myofascial pain.	[5]
Kasahara Y, et al.	2012	Japan	A telerobotic-assisted drilling system	Oral implantology	Cutting force transmits from a cutting material to a surgeon via the master–slave system.	It achieved precise manipulation of the drill feed and vivid feedback from the cutting force.	[6]
Jiang J G, et al.	2013	China	A dental arch generator with a hardware control scheme based on the industrial personal computer and control card PC6401	Prosthodontics-tooth arrangement	The dental arch generator can automatically generate a dental arch to fit a patient according to the patient’s arch parameters.	The system can be used to fabricate full dentures and bend orthodontic wires.	[7]
Jin-gang J, et al.	2013	China	An archwire bending robot	Orthodontics	The overall structure, bend die and archwire supporting part of archwire bending robot for orthodontic treatment is designed.	This paper proposes to use robots to replace dentists for completing orthodontic wires bending.	[8]
Wang D, et al.	2014	China	LaserBot	Prosthodontics-tooth preparation	It can manipulate a femtosecond laser beam to drill/burr a decayed tooth to realize clinical tooth crown preparation	It achieved precise 3D motion control of a laser focal point and is small enough to be used in the narrow workspace of the oral cavity.	[9]
Sun X, et al.	2014	USA	A robotic system for automated site preparation for dental implants	Oral implantology	Phantom experiments proved that the complicated volumes of the natural-root-formed implants can be accomplished.	With this robotic system, controlled and accurate drilling was achieved, which made more advanced implant designs possible.	[10]
Toosi A, et al.	2014	Iran	A haptic virtual reality simulator for root canal treatment	Endodontics	The user can burr the enamel and dentin until reaching the pulp chamber and then clean the internal surface of a root canal using a simulated K-file.	It helps improving the training available in the field of endodontics.	[11]
Razavi M, et al.	2015	Iran	A haptics- based tooth drilling simulator	Endodontics	The proposed idea for force calculation leads to a uniform sensation of force	An important feature of the designed system is the capability to run in a real-time fashion.	[12]
Wang G, et al.	2015	China	A novel biomimetic chewing robot	Other-Masticatory	The chewing robot is able to simulate the motion of human mastication in a biologically faithful way.	Two higher kinematic pairs of point contact are proposed to simulate the two temporomandibular joints.	[13]
Yuan F, et al.	2016	China	An automatic tooth preparation robotic device with three-dimensional motion planning software	Prosthodontics-tooth preparation	The results validated the accuracy and feasibility of the automatic tooth preparation technique	The results illustrated the potential of the automatic tooth preparation technique for use in dental clinics.	[14]
Chao A H, et al.	2016	USA	KUKA, Augsburgs, Germany	Oral surgery	This preclinical study demonstrates the feasibility of pre-programmed robotic osteotomies for free fibula flap mandible reconstruction.	This method exhibits high degrees of linear and angular accuracy, and may be of utility in the development of techniques to further improve surgical accuracy.	[15]
Zhu J H, et al.	2016	China	An automatical custom three-arm robotic system	Oral surgery	The accuracy of the fibula implant orientation with the robotic system was comparable to that with navigation system and superior to that with the freehand technique.	The robotic system is feasible, efficient, and reliable for mandibular reconstruction.	[16]
Sakaeda G, et al.	2017	Japan	An automatic teeth cleaning mouthpiece robot	Other-tooth cleaning	The robot system was operated correctly during a demonstration test.	This robot was developed to support elderly and handicapped people who cannot brush their teeth without assistance from helpers.	[17]

## Data Availability

All relevant data are included within the paper itself.

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
