# Peer review of "Robotics in Dentistry: A Narrative Review"

_dentistry, 2023, doi:10.3390/dj11030062_

Round 1
Reviewer 1 Report
Dear authors, the topic is very intresting.
I suggest the following
1. Change of the title . It is not a systematic review thus you should mention "narrative" review.
2. In materials and methods: you should incorporate a PRISMA flow chart of your articles.
3. Important! You should fix a table with columns such as the authors, title, system they describe, category of the system (clinical application or laboratory application), company-country, results of testing, category of dentistry field.. etc. it will help readers to have an overview
Introduction and discusion (now named conclusion) are poorly described and not well documented. I suggest rewritting them and put conclusion in a different section. We miss your overview and your assessment of the question "where are we going? what will be the future?"
Thank you for your effort
Author Response
We are appreciative of the insightful comment provided by the reviewer. In response to the suggestion, the manuscript has undergone the following revisions.
Comment 1: Change of the title. It is not a systematic review thus you should mention "narrative" review.
Reply 1: The title of the manuscript has been altered in accordance with the reviewer's comment.
Comment 2: In materials and methods: you should incorporate a PRISMA flow chart of your articles.
Reply 2: A PRISMA flow chart has been incorporated as Figure 1.
Comment 3: Important! You should fix a table with columns such as the authors, title, system they describe, category of the system (clinical application or laboratory application), company-country, results of testing, category of dentistry field.. etc. it will help readers to have an overviewIntroduction and discusion (now named conclusion) are poorly described and not well documented. I suggest rewritting them and put conclusion in a different section. We miss your overview and your assessment of the question "where are we going? what will be the future?"
Reply3: A comprehensive summary of the principal articles and their respective details has been presented in Table 1. Moreover, the manuscript has been reorganized and revised, and a Discussion and Conclusion section has been added, which encompasses our perspectives on the future of robotics in dentistry.
Reviewer 2 Report
Dear Authors, your work is very interesting, but it requires significant correction:
1. title - please specify if this is a narrative or systematic review, although the rest of the text more suggests that this is a systematic review
2. material and methods - no detailed description of the inclusion and exclusion criteria - I suggest making a table, no PRISMA scheme, no PICO
3. Results - no risk of bias
4. Discussion - I suggest making a table summarizing the analyzed publications.
Author Response
We are grateful for the invaluable advice provided by the reviewer. In response to the suggestion, the manuscript has undergone the following revisions.
Comment 1: title - please specify if this is a narrative or systematic review, although the rest of the text more suggests that this is a systematic review
Reply 1: The title of the manuscript has been altered in accordance with the reviewer's comment.
Comment 2: material and methods - no detailed description of the inclusion and exclusion criteria - I suggest making a table, no PRISMA scheme, no PICO
Replay 2: Inclusion and exclusion criteria have been documented, and a PRISMA flow chart has been added as Figure 1.
Comment 3: Discussion - I suggest making a table summarizing the analyzed publications.
Reply 3: A comprehensive summary of the principal articles and their respective details has been presented in Table 1.
Reviewer 3 Report
DEAR AUTHORS, title of study is attractive but the manuscript has severe flaws.
why the PRISMA checklist Are not followed?
why is there no registration of review in any registration site ? eg PROSPERO
WHY THERE IS NOT FLOW CHART showing the progress of review?
even though you presented the use of robotics in various field but a table of your considered articles would have been enhanced the manuscript.
why not prevalence, perception knowledge evaluating articles not considered?
please look into these points re-write / add these aspects and resubmit.
Author Response
We are appreciative of the insightful comment provided by the reviewer. In response to the suggestion, the manuscript has undergone the following revisions.
Comment:
why the PRISMA checklist Are not followed?
why is there no registration of review in any registration site ? eg PROSPERO
WHY THERE IS NOT FLOW CHART showing the progress of review?
even though you presented the use of robotics in various field but a table of your considered articles would have been enhanced the manuscript.
why not prevalence, perception knowledge evaluating articles not considered?
please look into these points re-write / add these aspects and resubmit.
Reply:
- A PRISMA flow chart has been incorporated as Figure 1 and a comprehensive summary of the principal articles and their respective details has been presented in Table 1.
- The reviewer commented on why studies on prevalence and perception knowledge evaluation were not considered. The objective of this article is to provide an in-depth evaluation of the utilization and research advancements in the field of robotics in dentistry. Currently, the implementation of dental robotics in clinical practice remains limited due to the paucity of data. Given this, we believe that any analysis of its prevalence or evaluation of perception knowledge would be inherently subjective and one-sided. To objectively present the current state of dental robotics and avoid biasing the reader's subjective opinions and assessments, studies on prevalence and perception knowledge evaluation have been omitted.
- The manuscript has been thoroughly proofread by a native speaker at Cactus Communications Inc.'s 'editage' service.
Round 2
Reviewer 1 Report
Εffort to improve manuscript has been made. I would prefer chronologically presentation of articles in the table with the studies. Conclusions are better presented in the revised form. I expected though more in the discussion part.
Author Response
We express our gratitude for the invaluable comments you have provided. In response to the suggestion, the manuscript has undergone the following revisions:
1. Table 1 has undergone a modification to ensure the table of studies presented a chronological progression of articles.
2. The discussion section has been updated with the recent advancements, prospects, and issues of AI and robotics.
Reviewer 2 Report
Dear Authors, the current version is written correctly, the work is very interesting. Very nice, clear table. I have no objections.
Author Response
We express our gratitude for the invaluable comments you have provided, and we deeply appreciate the time and effort you have dedicated.
Reviewer 3 Report
congratulations...for your work
Author Response

(The authors gave the same response as above.)
